# CD14 release induced by P2X7 receptor restricts inflammation and increases survival during sepsis

Cristina Alarcón-Vila[1], Alberto Baroja-Mazo[1†], Carlos de Torre-Minguela[1†], Carlos M Martínez[2], Juan J Martínez-García[1‡], Helios Martínez-Banaclocha[1], Carlos García-Palenciano[3], Pablo Pelegrin[1*]

[1]Línea de Inflamación Molecular, Instituto Murciano de Investigación Biosanitaria IMIB-Arrixaca, Hospital Clínico Universitario Virgen de la Arrixaca, Murcia, Spain; [2]Plataforma de Patología, Instituto Murciano de Investigación Biosanitaria IMIB-Arrixaca, Murcia, Spain; [3]Unidad de Reanimación, Hospital Clínico Universitario Virgen de la Arrixaca, Murcia, Spain

**Abstract** P2X7 receptor activation induces the release of different cellular proteins, such as CD14, a glycosylphosphatidylinositol (GPI)-anchored protein to the plasma membrane important for LPS signaling via TLR4. Circulating CD14 has been found at elevated levels in sepsis, but the exact mechanism of CD14 release in sepsis has not been established. Here, we show for first time that P2X7 receptor induces the release of CD14 in extracellular vesicles, resulting in a net reduction in macrophage plasma membrane CD14 that functionally affects LPS, but not monophosphoryl lipid A, pro-inflammatory cytokine production. Also, we found that during a murine model of sepsis, P2X7 receptor activity is important for maintaining elevated levels of CD14 in biological fluids and a decrease in its activity results in higher bacterial load and exacerbated organ damage, ultimately leading to premature deaths. Our data reveal that P2X7 is a key receptor for helping to clear sepsis because it maintains elevated concentrations of circulating CD14 during infection.

*For correspondence:
pablo.pelegrin@imib.es

†These authors contributed equally to this work

Present address: ‡U1003 Phycell lab, Inflammasome and Ion channels, Université de Lille, INSERM, Institut Pasteur de Lille, Lille, France

Competing interests: The authors declare that no competing interests exist.

## Introduction

Purinergic signaling controls many different processes during infection and inflammation (*Eltzschig et al., 2012*) and the P2X7 receptor is one of the key purinergic receptors in modulating the macrophage functions that orchestrate the inflammatory response (*Di Virgilio et al., 2017*). The P2X7 receptor in LPS-primed macrophages activates the nucleotide-binding domain and the leucine-rich repeat receptor pyrin domain containing 3 (NLRP3) inflammasome, which in turn leads to the release of pro-inflammatory cytokines from the interleukin (IL)−1 family, such as IL-1β (*Di Virgilio et al., 2017*). However, the P2X7 receptor can also block NLRP3 if it is activated before the LPS priming of the macrophages (*Martínez-García et al., 2019*). This means that the P2X7 receptor can cause different pro- or anti-inflammatory responses depending when it is activated. In macrophages, the P2X7 receptor also induces the release of extracellular vesicles that contain IL-1β and MHCII (*MacKenzie et al., 2001*; *Pellegatti et al., 2008*; *Qu et al., 2009*; *Qu et al., 2007*); however, apart from these proteins, the cargo of P2X7 receptor-induced extracellular vesicles, remains largely unknown. A large fraction of IL-1β and other cytosolic proteins are also released via pyroptosis, a type of cell death dependent on inflammasome activation and the formation of large plasma membrane pores by gasdermin D (*Broz et al., 2020*). The secretome of the P2X7 receptor in macrophages includes many soluble proteins released by pyroptosis and some prototypic proteins of extracellular vesicles, such as annexin A1 (*de Torre-Minguela et al., 2016*). One of the plasma membrane-associated proteins identified as part of the P2X7 receptor secretome is CD14, a well-known

**eLife digest** When the immune system detects an infection, it often launches an inflammatory response to fight off the disease. This defense mechanism is activated by a cascade of signaling molecules that can aggravate inflammation, causing it to damage the body's own tissues and organs. This life-threatening reaction is referred to as sepsis, and kills around 11 million people each year. New approaches are therefore needed to help alleviate the damage caused by this condition.

The inflammatory response is often triggered by proteins called receptors, which sit on the surface of immune cells. When these receptors are activated, they induce cells to secrete proteins that travel around the body and activate immune cells that can eliminate the infection. In 2016, a group of researchers showed that a receptor called P2X7 stimulates the release of a signaling molecule called CD14. Patients with sepsis often have elevated amounts of CD14 in their bloodstream. Yet, it remained unclear what causes this rise in CD14 and what role this molecule plays in the development of sepsis.

Now, Alarcón-Vila et al. – including some of the researchers involved in the 2016 study – have investigated the role of P2X7 in mice undergoing sepsis. This was done by puncturing the mice's intestines, causing bacteria to leak out and initiate an over-active immune response. Alarcón-Vila et al. found that mice lacking the P2X7 receptor had less CD14 and struggled to eliminate the bacterial infection from their system. This increase in bacteria caused excessive damage to the mice's organs, ultimately leading to premature death.

These findings suggest that P2X7 plays an important role in preventing the onset of sepsis by helping maintain high levels of CD14 following infection. This result could help to identify new therapies that reduce the mortality rates of septic infections.

myeloid cell marker (*de Torre-Minguela et al., 2016*; *Setoguchi et al., 1989*). CD14 is a glycosyl-phosphatidylinositol (GPI)-anchored membrane protein important for transferring LPS to Toll-like receptor (TLR) 4 and for controlling TLR4 translocation to endosomes that activate TRAM-TRIF-dependent pathways (*Zanoni and Granucci, 2013*). Therefore, CD14 is important for ensuring that TLR4 responds optimally to LPS and that macrophages produce pro-inflammatory cytokines. CD14 could be released from the cells by the action of proteinase (*Wu et al., 2019*); however, there are also different release mechanisms independent of proteinase that are not known. The pool of extracellular CD14 in vivo, known as soluble CD14, is detected in different fluids during infection and particularly during sepsis, a life-threating condition resulting from exacerbated inflammation in response to infection (*Barratt-Due et al., 2017*; *Bas et al., 2004*). Extracellular CD14 during microbial infection is important for host defense, in particular for bacterial clearance, but little is known about the release of CD14 in vivo (*Qin et al., 2017*; *Knapp et al., 2006*; *Sahay et al., 2018*; *Wieland et al., 2005*). Despite the fact that the P2X7 receptor induces the release of CD14 (*de Torre-Minguela et al., 2016*), it is not known if P2X7 contributes to the extracellular pool of CD14 during infection or what its role is in defending the host during sepsis. In this study, we found for first time that CD14 is a cargo of extracellular vesicles released by P2X7 receptor activation and functionally that the lack of cellular CD14 compromises the production of macrophage pro-inflammatory cytokines. Additionally, we also found that during sepsis there is a decrease in the extracellular pool of CD14 in $P2rx7^{-/-}$ mice, which results in high bacterial dissemination and a decreased mice survival and reveals that the P2X7 receptor is important for maintaining an optimum level of CD14 and thus ensuring survival of sepsis.

## Results

### P2X7 receptor stimulation induces the release of extracellular vesicles containing CD14

In a previous study, we identified CD14 as a specific component of the P2X7 receptor secretome in macrophages (*de Torre-Minguela et al., 2016*). After ATP stimulation of LPS-primed macrophages, we detected the appearance of extracellular CD14 in the 100K pellet together with the tetraspanin CD9, a well-known extracellular vesicle marker (*Figure 1a*). This pellet contained extracellular

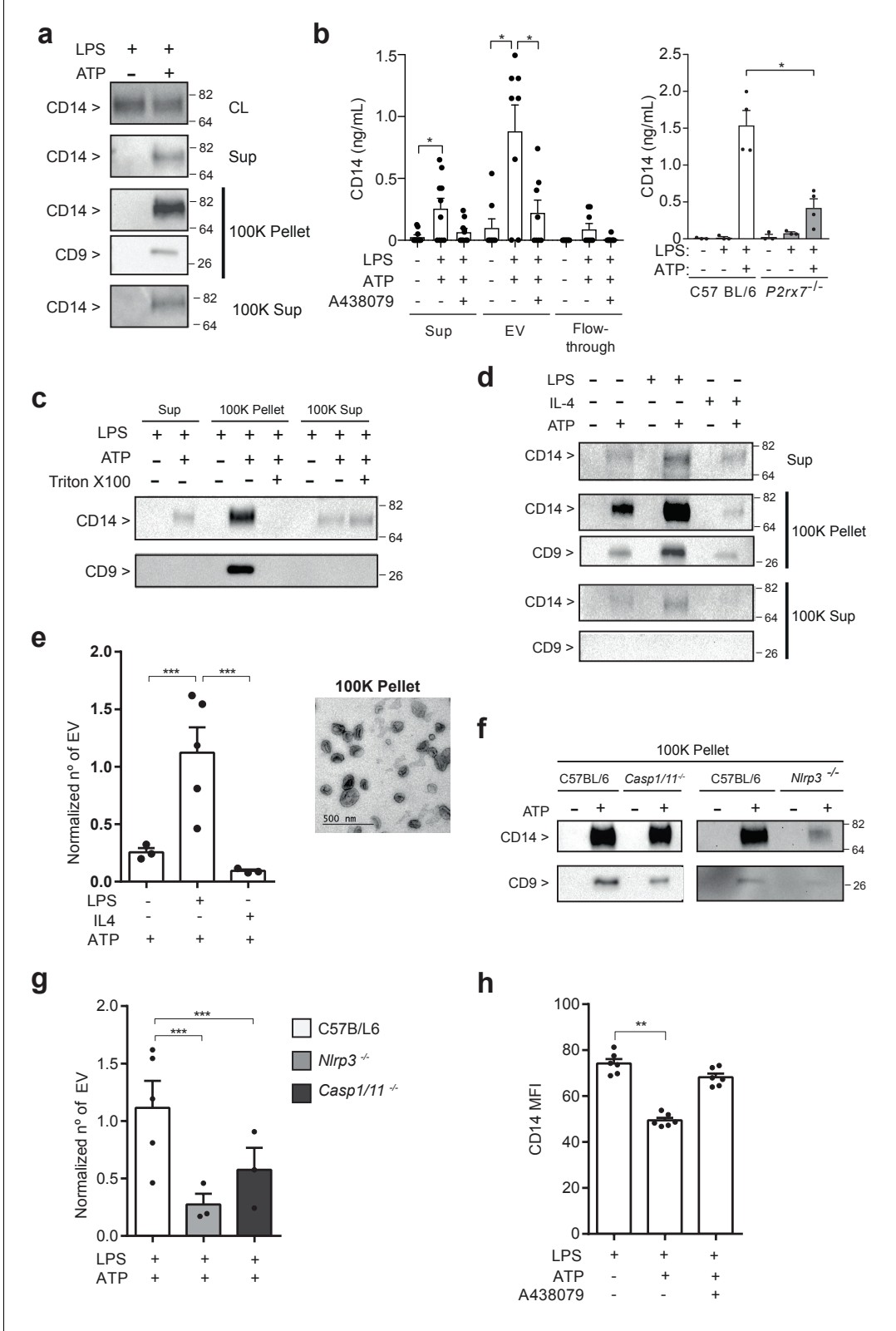

**Figure 1.** P2X7 receptor stimulation induces the release of extracellular vesicles containing CD14. (a) Immunoblot for CD14 and CD9 in cell lysate (CL), cell-free supernatant (Sup) and supernatant fractions (100K pellet and 100K supernatant) obtained from extracellular vesicle (EV) isolation from BMDMs treated for 4 hr with LPS (10 ng/ml) and then stimulated or not for 20 min with ATP (3 mM); representative of $n$ = 3 experiments. (b) Quantification of extracellular CD14 by ELISA in Sup, EV isolated with the Exo-Quick kit and flow-through fraction obtained in cell-free supernatants from BMDM treated

*Figure 1 continued on next page*

*Figure 1 continued*

as in (**a**), but before ATP application cells were treated for 10 min with A438079 (20 µM) as indicated (left) or from EV isolated with the Exo-Quick in supernatants from *P2rx7*$^{-/-}$ macrophages (right); each dot represents an independent experiment (*n* = 4 to 8). (**c**) Immunoblot for CD14 and CD9 in Sup, 100K pellet and 100K supernatant from BMDM cell-free supernatants treated as in (**a**), but after the first step of EV isolation, Sup was treated with 2% of Triton X-100; representative of *n* = 3 independent experiments. (**d**) Immunoblot for CD14 and CD9 in Sup, 100K pellet and 100K supernatant in cell-free supernatants from BMDM unprimed or primed for 4 hr with LPS (10 ng/ml) or IL-4 (20 ng/ml) and then treated with ATP as in (**a**); representative of *n* = 3 experiments. (**e**) Quantification of EV released from BMDM treated as in (**d**), left panel; each dot represents an independent experiment (*n* = 3 to 5); Normalized number of EV to the number of cells obtained in each treatment is shown. Representative transmission electron microscopy image obtained from the 100K pellet, right panel. (**f**) Immunoblot for CD14 and CD9 in 100K pellet obtained from cell-free supernatants of C57BL/6 (wild-type), *Nlrp3*$^{-/-}$ or *Casp1/11*$^{-/-}$ BMDM treated as in (**a**), representative of *n* = 3 independent experiments. (**g**) Quantification of EV in cell-free supernatants of C57BL/6 (wild-type), *Nlrp3*$^{-/-}$ or *Casp1/11*$^{-/-}$ BMDM treated as in (**a**); each dot represents an independent experiment (*n* = 3 to 5); Normalized number of EV to the number of cells obtained in each treatment is shown. (**h**) Quantification of CD14 mean fluorescence intensity (MFI) in BDMD treated as in (**b**); each dot represents an independent experiment (*n* = 6). *p<0.05, **p<0.01, ***p<0.001, Mann-Whitney test. For a, c, d, and f numbers on the right of the blots correspond to the molecular weight in kDa.

The online version of this article includes the following source data and figure supplement(s) for figure 1:

**Source data 1.** Source data file for *Figure 1*.
**Figure supplement 1.** Characterization of P2X7 receptor-induced extracellular vesicles.
**Figure supplement 2.** Electron microscopy of P2X7 receptor-induced extracellular vesicles.

vesicles with an average size of 167.4 nm (*Figure 1—figure supplement 1a*). Most of the P2X7 receptor secretome examined, with the exception of IL-1β, was present in the soluble fraction and not associated with the 100K pellet (*Figure 1—figure supplement 1b*). The presence of CD14 in extracellular vesicles obtained from macrophage supernatants after P2X7 receptor stimulation was also determined by using a protocol for extracellular vesicle isolation based in polymer-precipitation (*Figure 1b*). The treatment of whole cellular supernatants with Triton X100 before vesicle isolation resulted in a loss of CD14 from the 100K pellet fraction that was detected in the soluble fraction (100K supernatant) (*Figure 1c*), suggesting CD14 is a cargo of the vesicles. The CD14 in the 100K pellet was detected just after 5 min of ATP stimulation (*Figure 1—figure supplement 1c*) and was independent of the macrophage activation polarity, as it was detected from M1 and M2 macrophages (*Figure 1d*). However the number of extracellular vesicles released in LPS-primed macrophages (M1) after P2X7 receptor stimulation was significantly higher than those released from IL-4 treated (M2) or resting macrophages (*Figure 1e* and *Figure 1—figure supplement 2a*). Interestingly, the release of CD14 induced by the P2X7 receptor in extracellular vesicles was highly dependent on the NLRP3 inflammasome, but not on caspase-1 (*Figure 1f*). Then, we found that the amount of extracellular vesicles released after activation of the P2X7 receptor in *Nlrp3*$^{-/-}$ and *Casp1/11*$^{-/-}$ macrophages was smaller when compared with wild-type macrophages (*Figure 1g* and *Figure 1—figure supplement 2b*). The morphology of *Nlrp3*$^{-/-}$-derived extracellular vesicles was similar to that of resting or IL-4-primed wild-type macrophages (*Figure 1—figure supplement 2a, b*). In contrast, *Casp1/11*$^{-/-}$-derived extracellular vesicles were similar in morphology to LPS-primed wild-type macrophages (*Figure 1—figure supplement 2a,b*), indicating that a specific pool of extracellular vesicle dependent on LPS-priming and NLRP3 could be enriched in CD14 and explain the differences of released CD14 found among *Nlrp3*$^{-/-}$ and *Casp1/11*$^{-/-}$ macrophages. The release of CD14 observed in ATP-treated macrophages resulted in a significant decrease in cell surface CD14 (*Figure 1h*), thus suggesting that the P2X7 receptor could induce a decrease in CD14-dependent signaling in macrophages as well as being a source of extracellular CD14.

## P2X7 receptor stimulation impairs LPS-mediated signaling

CD14 is a co-receptor of TLR4 important for LPS signaling (*Zanoni and Granucci, 2013*), so we investigated whether CD14 released after P2X7 receptor activation affect the signaling of LPS in macrophages. Treatment of macrophages with extracellular ATP and then subjected to LPS activation resulted in a decrease in LPS-induced expression and a secretion of the pro-inflammatory cytokines IL-6 and TNF-α (*Figure 2a,b*). This effect was abrogated when the specific P2X7 receptor antagonist A438079 was used or when macrophages were isolated from *P2rx7*$^{-/-}$ mice (*Figure 2a, b*), thus suggesting that the P2X7 receptor was affecting LPS signaling in macrophages. LPS is not able to activate TLR4 in the absence of CD14 at the cell membrane (*Pizzuto et al., 2018*). By

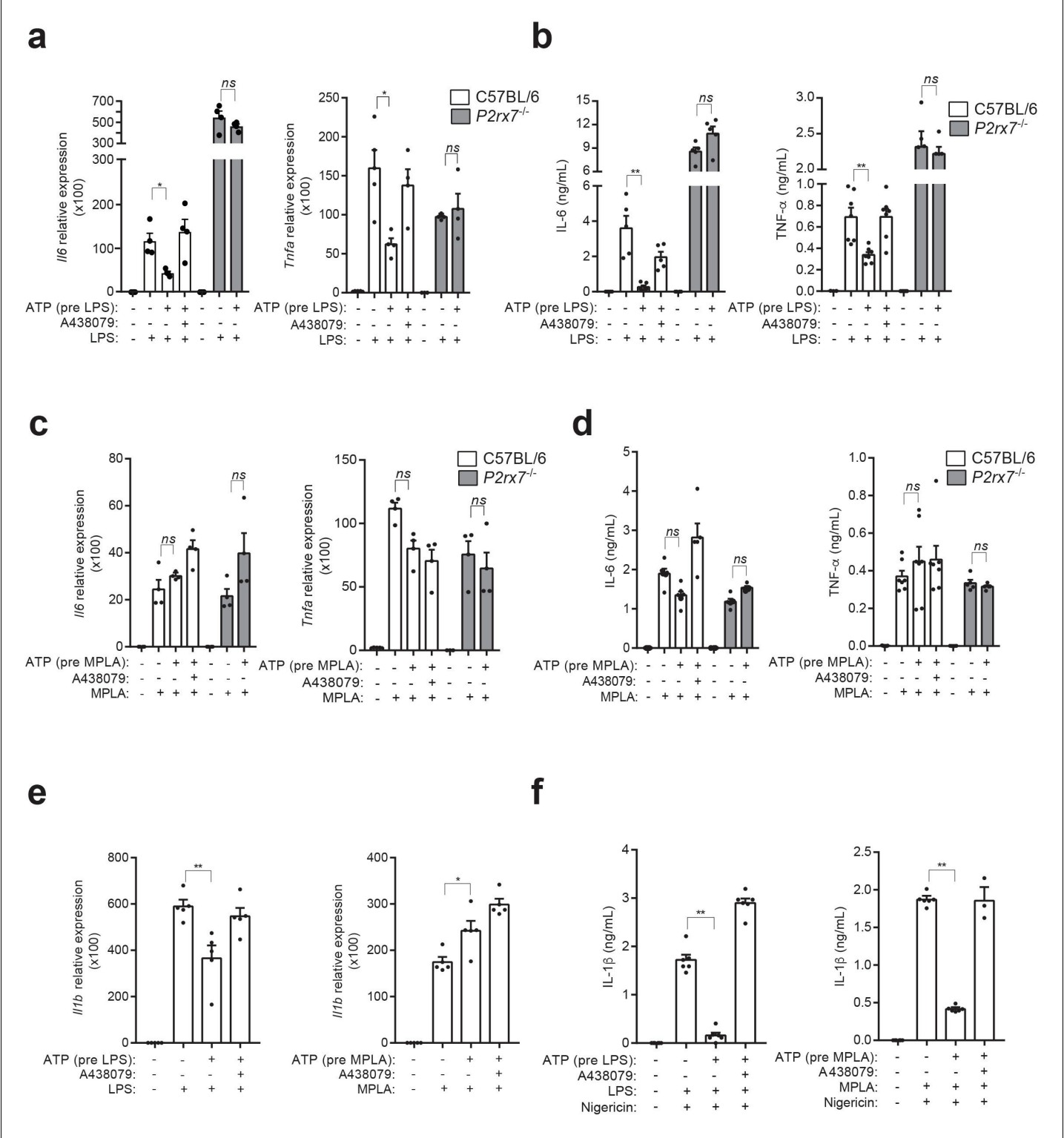

**Figure 2.** P2X7 receptor stimulation impairs LPS-mediated signaling. (a) Expression of *Il6* and *Tnfa* genes analyzed by qPCR in C57BL/6 (wild-type) or *P2rx7*$^{-/-}$ BMDM treated or not for 10 min with A438079 (10 μM), then incubated for 30 min with ATP (5 mM), then washed and finally primed for 4 hr with LPS (10 ng/ml). (b) IL-6 and TNF-α concentration in cell-free supernatants from C57BL/6 (wild-type) or *P2rx7*$^{-/-}$ BMDM treated as in (a). (c,d) Expression of *Il6* and *Tnfa* genes analyzed by qPCR (c) and ELISA for IL-6 and TNF-α in cell-free supernatants (d) from C57BL/6 (wild-type) or *P2rx7*$^{-/-}$ BMDM treated as in (a) but finally stimulated for 4 hr with MPLA (1 μg/ml) instead of LPS. (e) Expression of *Il1b* gene analyzed by qPCR from BMDM treated as in (a) and (c). (f) IL-1β concentration in cell-free supernatants from BMDM treated as in (e) and after LPS or MPLA stimulation, cells were

*Figure 2 continued on next page*

*Figure 2 continued*

incubated for 30 min with nigericin (10 μM). Each dot represents a single independent experiment; data are represented as mean ± SEM; *n* = 4–6 single experiments; *p<0.05; **p<0.01; *ns*, no significant difference (p>0.05); Mann–Whitney test.

The online version of this article includes the following source data and figure supplement(s) for figure 2:

**Source data 1.** Source data file for *Figure 2*.

**Figure supplement 1.** Anti-CD14 reduced LPS, but not MPLA, induced cytokine production.

contrast, lipopolysaccharides with smaller hydrophilic moiety, like monophosphoryl lipid A (MPLA), signals independently of CD14 (*Maeshima and Fernandez, 2013*). In order to investigate the role of P2X7 receptor-induced CD14 release in the reduction of LPS signaling, we tested whether also MPLA signaling was impaired by P2X7 receptor activation. When macrophages were stimulated with MPLA, the production of IL-6 and TNF-α was not affected when P2X7 receptor was activated (*Figure 2c,d*). This effect was similar when CD14 was blocked with a specific antibody; when this was done, it decreased the production of both cytokines after LPS was added to stimulate the macrophages but not when MPLA was used (*Figure 2—figure supplement 1a,b*). Our group has recently reported that in macrophages, activating the P2X7 receptor before LPS-priming also inhibits NLRP3 inflammasome, a phenomena mediated by P2X7 receptor-mediated mitochondrial damage (*Martínez-García et al., 2019*). To assess the possible involvement of CD14 release in NLRP3 inhibition, we first measured *Il1b* gene expression and found that, similarly to *Il6* and *Tnfa*, ATP treatment decreased the expression of *Il1b* when LPS was used to activate the cells, but not when MPLA was used (*Figure 2e*), suggesting that the decrease of CD14 on cell membrane affects the priming by LPS. However, the release of IL-1β induced by nigericin decreased when macrophages were incubated with ATP before LPS- or MPLA-priming (*Figure 2f*), which suggests that *Il1b* production, but not NLRP3 activation, was affected by the release of CD14 induced by the activation of the P2X7 receptor.

## P2X7 receptor controls CD14 in extracellular vesicles during sepsis

The extracellular pool of CD14 increases during infection and sepsis (*Bas et al., 2004*), and here we confirm that septic patients presented elevated levels of CD14 in the plasma when compared to non-septic volunteers (*Figure 3a* and *Supplementary file 1*-Table 1). Similarly, the presence of P2X7 receptor in monocytes was also elevated in septic individuals (*Figure 3a*), in accordance to previous studies (*Martínez-García et al., 2019*). To study if P2X7 receptor is important in maintaining the extracellular pool of CD14 during infection, we performed the cecal ligation and puncture (CLP) procedure in wild-type and *P2rx7−/−* mice and we found that the lack of P2X7 receptor expression resulted in reduced levels of cell-free CD14 in both serum and peritoneal lavage (*Figure 3b*). Similarly, administration of the specific P2X7 receptor-antagonist A438079 to wild-type mice subjected to CLP resulted in a reduction in CD14 in the peritoneal lavage and a mild reduction in serum (*Figure 3c*). This result could be probably due to the site of A438079 injection, which was i.p., and to its short half-life, which hampers its ability to reach the blood serum (*McGaraughty et al., 2007*). Furthermore, the increase in CD14 detected in the peritoneal lavage was mainly associated with the extracellular vesicle pool (*Figure 3d*), and the presence of CD14 in extracellular vesicles decreased in P2X7 receptor deficient mice (*Figure 3d*). These data suggest that during infection, the P2X7 receptor contribute to the presence of extracellular CD14 in extracellular vesicles.

## Deficiency in the P2X7 receptor increases cytokine production during sepsis

In order to determine if the P2X7 receptor-dependent release of CD14 during sepsis was impairing LPS-signaling in vivo, we measured cytokines in the serum of P2X7 deficient mice subjected to CLP. In line with our in vitro data, levels of IL-6 appeared higher in the *P2rx7−/−* mice after CLP compared to wild-type (*Figure 4a*). This was also confirmed for other cytokines, chemokines and acute phase proteins measured in the serum of *P2rx7−/−* mice as well as in wild-type mice treated with A438079 (*Figure 4b,c*), which suggests that P2X7 receptor is important for the downregulation of cytokines during sepsis.

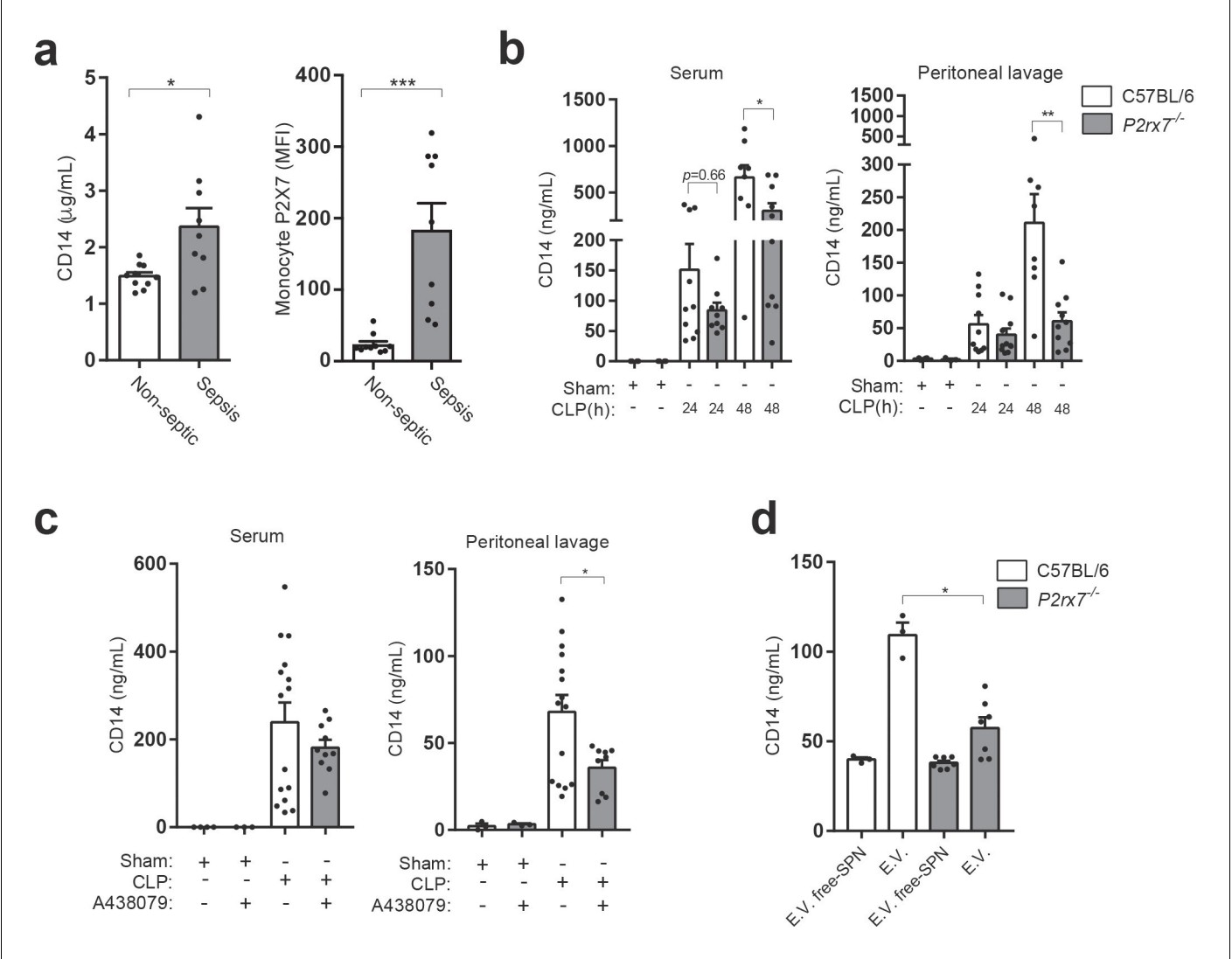

**Figure 3.** P2X7 receptor controls CD14 in extracellular vesicles during sepsis. (a) Blood plasma concentration of CD14 (left) and quantification of P2X7 receptor mean fluorescence intensity (MFI) in monocytes (right) from non-septic donors and intra-abdominal origin septic patients within the first 24 hr of admission to the surgical unit. Each dot represents a donor or septic individual, n = 10. (b) CD14 concentration in the serum and peritoneal lavage of C57BL/6 (wild-type) and $P2rx7^{-/-}$ mice collected 24 and 48 hr after CLP measured by ELISA. (c) CD14 concentration in the serum and peritoneal lavage of C57BL/6 (wild-type) mice collected 24 hr after CLP, treated or not with A438079 (100 µM/kg) 1 hr before CLP. (d) CD14 concentration in extracellular vesicles (E.V.) isolated from the peritoneal lavage of C57BL/6 (wild-type) and $P2rx7^{-/-}$ mice collected 48 hr after CLP. For b–d, each dot represents a single mouse; data are represented as mean ± SEM; *p<0.05; **p<0.01; ***p<0.0001; Mann-Whitney test.
The online version of this article includes the following source data for figure 3:

**Source data 1.** Source data file for *Figure 3*.

## Extracellular CD14 induced by the P2X7 receptor during sepsis controls bacterial dissemination and cytokine secretion

In order to elucidate if extracellular CD14 released by P2X7 has a role during sepsis, we next analyzed bacterial dissemination, as it is known that extracellular CD14 contributes to the clearance of invading bacteria and that the P2X7 receptor is important for controlling bacterial content during sepsis (*Csóka et al., 2015*; *Lévêque et al., 2017*). As expected, the bacterial burden increased in serum, peritoneal cavity and liver in $P2rx7^{-/-}$ compared to wild-type mice after CLP (*Figure 5a*). Similarly, wild-type mice treated with the P2X7 receptor-antagonist A438079 also presented an increase in bacterial load (*Figure 5b*). Administration of recombinant CD14 to $P2rx7^{-/-}$ mice before the CLP procedure resulted in a significant reduction in bacterial load in the serum, peritoneal lavage

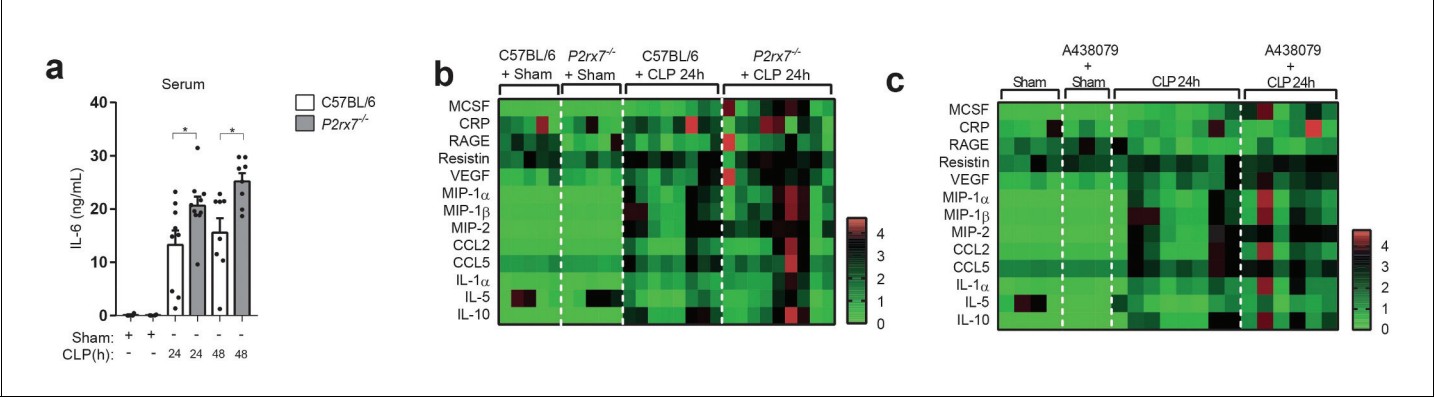

**Figure 4.** The deficiency or blocking of P2X7 receptor increases cytokine production during sepsis. (**a**) ELISA of IL-6 in the serum of C57BL/6 (wild-type) and *P2rx7*⁻/⁻ mice collected 24 and 48 hr after CLP; each dot represents a single mouse; data are represented as mean ± SEM; *p<0.05; Mann-Whitney test. (**b,c**) Heatmaps for the concentrations of different cytokines, chemokines and acute phase proteins as indicated in the serum of C57BL/6 (wild-type) and *P2rx7*⁻/⁻ mice (**b**) or C57BL/6 treated with A438074 (100 µM/kg) (**c**) collected 24 hr after CLP. For (**b,c**) C57BL/6 sham *n = 5* (**b**) and *n = 4* (**c**); *P2rx7*⁻/⁻ sham *n = 5*; sham+A438079 *n = 3*; C57BL/6 CLP *n = 8* (**b**) and *n = 8* (**c**); *P2rx7*⁻/⁻ CLP *n = 9*; and C57BL/6 CLP+A438079 *n = 6*. The online version of this article includes the following source data for figure 4:

**Source data 1.** Source data file for *Figure 4*.

and liver (*Figure 5c*). Cytokine and chemokine levels were reduced in the serum of *P2rx7*⁻/⁻ mice after CLP and treatment with recombinant CD14 (*Figure 5d,e*). These results suggest that extracellular CD14 is an important element in the P2X7 receptor secretome to control bacterial dissemination and cytokine production during sepsis.

## Release of P2X7 receptor-dependent CD14 during sepsis is important for survival

We and others have found that *P2rx7*⁻/⁻ mice present higher mortality during sepsis (*Csóka et al., 2015*; *Martínez-García et al., 2019*), and here we also confirm that treating wild-type mice with the pharmacological P2X7 receptor-antagonist A438079 before CLP also increases the likelihood of mortality (*Figure 6a*). In line, CLP resulted in higher decrease of mice body weight and poor well-being score on the supervision protocol in *P2rx7*⁻/⁻ mice and in wild-type mice treated with A438079 (*Figure 6—figure supplement 1a,b*). To test if the deficiency in extracellular CD14 in *P2rx7*⁻/⁻ mice during sepsis would be detrimental for survival, we treated *P2rx7*⁻/⁻ mice with recombinant CD14 before CLP and found that mice survival was significantly increased (*Figure 6b*), as well as weight loss was preserved (*Figure 6—figure supplement 1a*). This is in agreement with the reduction in the bacterial load induced by recombinant CD14 treatment (*Figure 5c*). We next evaluated organ damage as a direct cause of sepsis mortality and we found that the liver of wild-type mice displayed an unstructured parenchyma and ballooning hepatocytes after CLP (*Figure 6c* and *Supplementary file 1*-Table 2), which was aggravated in *P2rx7*⁻/⁻ mice or wild-type mice treated with A437089 and which further presented prominent steatosis and dying hepatocytes (*Figure 6c*). Spleen and lung damage induced by CLP in *P2rx7*⁻/⁻ mice or wild-type mice treated with A438079 were also exacerbated (*Figure 6—figure supplements 1c* and *2* and *Supplementary file 1*-Table 2). The spleen exhibited a severe depletion of white pulp with the presence of apoptotic bodies and congestion of red pulp (*Figure 6—figure supplements 1c* and *2*), whereas the lungs showed a marked leukocyte infiltration and intra-alveolar capillary hemorrhages with alveolar thickening (*Figure 6—figure supplements 1c* and *2*). Treating *P2rx7*⁻/⁻ mice with recombinant CD14 before CLP resulted in less pronounced damage in liver, spleen, and lung (*Figure 6d* and *Figure 6—figure supplement 3* and *Supplementary file 1*-Table 2). Altogether, our results demonstrate that the P2X7 receptor controls the release of CD14 in extracellular vesicles, impairing LPS signaling in myeloid cells and controlling bacteria and cytokine production during sepsis, thus reducing tissue damage and improving survival.

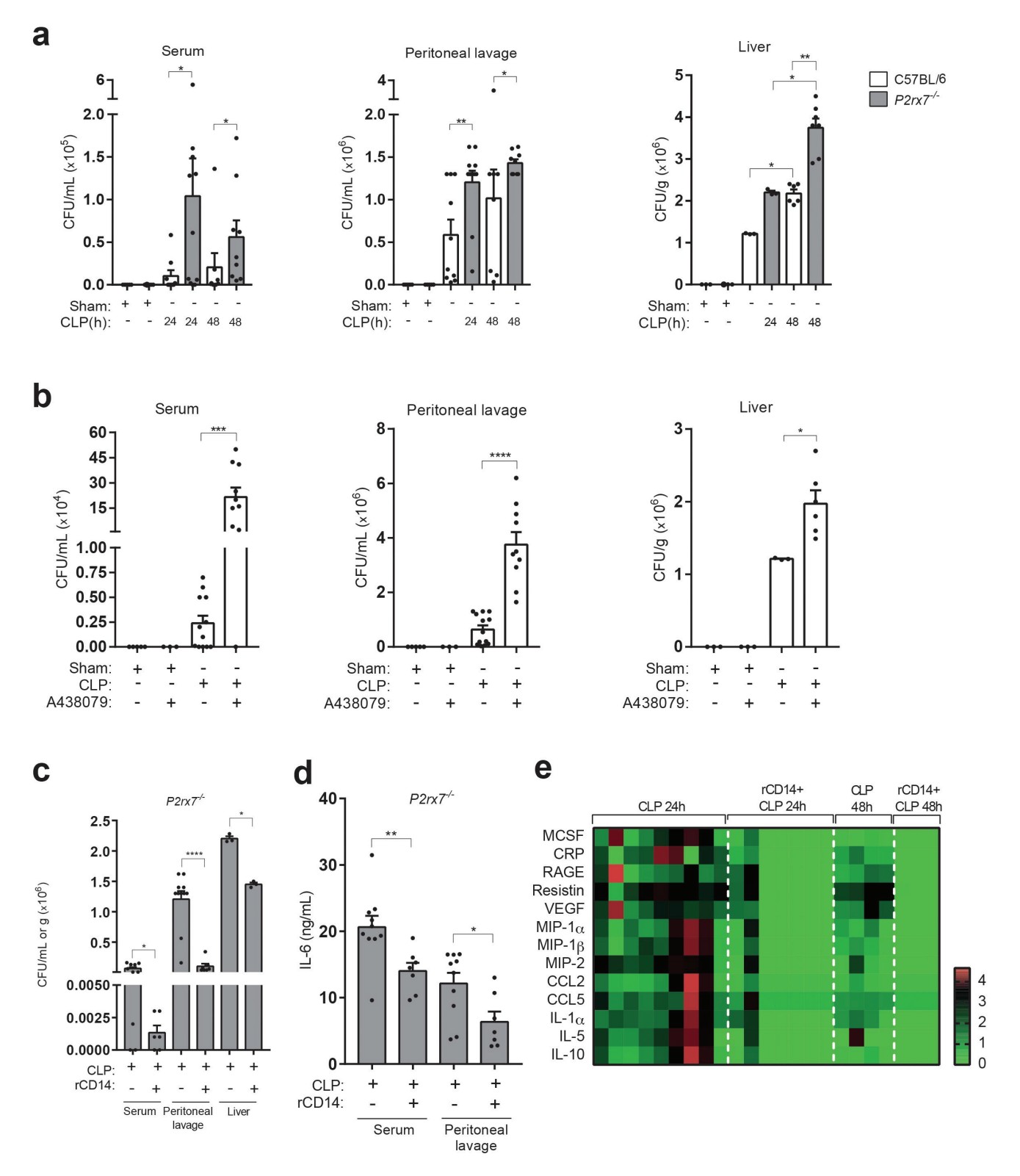

**Figure 5.** Extracellular CD14 limits bacterial dissemination and cytokine production during sepsis caused by P2X7 receptor deficiency. (a) Bacterial load in serum, peritoneal lavage and liver homogenates from C57BL/6 (wild-type) and *P2rx7*⁻/⁻ mice collected 24 and 48 hr after CLP. (b) Bacterial load in serum, peritoneal lavage, and liver homogenates from C57BL/6 (wild-type) mice treated with A438074 (100 μM/kg) and collected 24 hr after CLP. (c) Bacterial load in serum, peritoneal lavage and liver from *P2rx7*⁻/⁻ mice treated with recombinant CD14 (rCD14, 10 μg/g) 30 min before CLP and

*Figure 5 continued on next page*

*Figure 5 continued*

collected 24 hr after CLP. (**d**) ELISA for IL-6 in serum and peritoneal lavage samples from *P2rx7*⁻/⁻ mice collected 24 hr after CLP with or without treatment with recombinant CD14 (rCD14, 10 µg/g) 30 min before CLP; each dot represents a single mouse and data are represented as mean ± SEM. (**e**) Heatmaps for the concentrations of different cytokines, chemokines and acute phase proteins as indicated in the serum of *P2rx7*⁻/⁻ mice treated with rCD14 as in (**d**) collected 24 and 48 hr after CLP. For a-d panels, each dot represents a single mouse and data are represented as mean ± SEM; *p<0.05; **p<0.01; ***p<0.001; ****p<0.0001; Mann-Whitney test.

The online version of this article includes the following source data for figure 5:

**Source data 1.** Source data file for *Figure 5*.

## Discussion

Our study shows for first time that the cellular release of CD14 induced by the P2X7 receptor has two functional effects on the innate immune system: (*i*) it decreases CD14-dependent pro-inflammatory signaling in macrophages and (*ii*) it decrease bacterial dissemination, improving survival during sepsis. In macrophages, the activation of the P2X7 receptor controls many different responses, including the activation of the NLRP3 inflammasome or the unconventional release of different cellular proteins (*de Torre-Minguela et al., 2016*; *Di Virgilio et al., 2017*). The release mechanism for the secretome associated with P2X7 receptor activation is not characterized, but some are proteins released mainly by inflammasome-dependent pyroptosis (*de Torre-Minguela et al., 2016*). In this study, we describe that the release of CD14 correlates with the extracellular vesicle fraction, together with the tetraspanin CD9, rather than with the caspase-1 dependent pyroptotic soluble fraction. In fact, the release of CD14 was independent on caspase-1 activity. Due to the heterogeneity of extracellular vesicle populations released from cells and the fact that the P2X7 receptor has been associated with the release of different extracellular vesicles such as microvesicles and exosomes (*Kowal et al., 2016*; *MacKenzie et al., 2001*; *Qu et al., 2009*), our data supports that CD14 could be mainly a component of exosomes because CD14 largely appears associated with the high-speed pellet containing 'small' vesicles of ~160 nm. However, we could not rule out CD14 containing extracellular vesicles and exosomes may originate in the plasma membrane because there is a net reduction in plasma-membrane-associated CD14 and the presence of CD9 or IL-1β in this fraction has been correlated with both exosomes and plasma membrane-derived 'small' vesicles (*Kowal et al., 2016*; *MacKenzie et al., 2001*; *Qu et al., 2007*). Furthermore, the inflammasome deficiency does not affect the release of CD14, but it does decrease the amount of 'small' extracellular vesicles after P2X7 receptor activation, being this finding in accordance with a previous study demonstrating that NLRP3 impairs the release of exosomes from P2X7 receptor-activated dendritic cells (*Qu et al., 2009*). Therefore, it remains difficult to determine the exact nature of the extracellular vesicles containing CD14.

The release of CD14 occurred 5 min after stimulation in resting macrophages after brief activation of the P2X7 receptor. Under these circumstances, the NLRP3 inflammasome is not primed and therefore is not activated, thus protecting the cells from pyroptotic cell death (*Broz et al., 2020*). However, it has been recently described that prolonged P2X7 receptor activation would lead to apoptosis in resting macrophages (*Bidula et al., 2019*). The brief P2X7 receptor activation in resting macrophages with millimolar concentrations of ATP used in this study did not compromise cell viability and may resemble physiological conditions where ectonucleotidases provoke a fast ATP degradation in the extracellular milieu (*Eltzschig et al., 2012*). Under these conditions, there is a reduction in the subsequent production of pro-inflammatory cytokines after a smooth LPS activation that requires CD14 to signal via the TLR4-MD2 complex (*Pizzuto et al., 2018*; *Ryu et al., 2017*). This suggests that the release of CD14 from macrophages impairs CD14 signaling, and probably also the translocation of TLR4 complex to endosomes, thus impairing TRAM-TRIF-dependent pathways (*Zanoni and Granucci, 2013*). However, cytokine production was not affected when macrophages were treated with MPLA after P2X7 receptor activation, because MLPA does not require CD14 to signal (*Jiang et al., 2005*; *Maeshima and Fernandez, 2013*). The reduction in cytokine production upon LPS-priming induced by initial P2X7 receptor activation in macrophages is additional to the effect we have described on the inflammasome activation (*Martínez-García et al., 2019*), because NLRP3 activation was affected when macrophages were primed using both LPS and MLPA. All these suggest that brief P2X7 receptor activation before LPS priming has a widespread inhibitory effect on

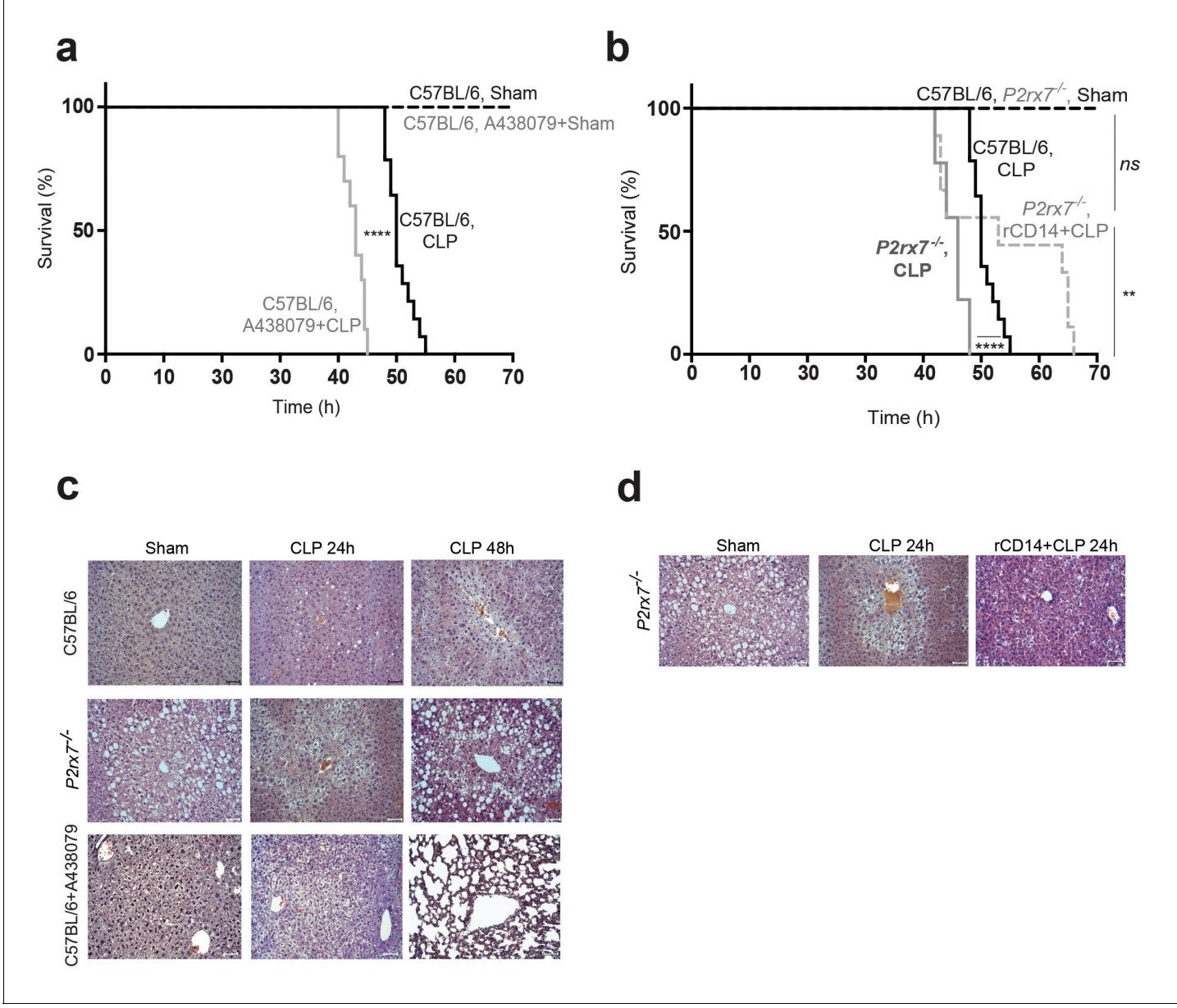

**Figure 6.** Release of P2X7 receptor-dependent CD14 during sepsis is important for survival. (a) Kaplan-Meier analysis of C57BL/6 (wild-type) mice survival after sham operation or CLP, a group of mice were treated with A438074 (100 μM/kg) before CLP. Sham groups n = 6 each; CLP n = 14 and CLP+A438079 n = 10. (b) Kaplan-Meier analysis of C57BL/6 (wild-type) and $P2rx7^{-/-}$ mice survival after sham operation or CLP. A group of $P2rx7^{-/-}$ mice were treated with recombinant CD14 (rCD14, 10 μg/g) 30 min before CLP. Sham groups n = 4 each; CLP n = 14, CLP $P2rx7^{-/-}$ n = 9; and rCD14+CLP $P2rx7^{-/-}$ n = 9. (c,d) Representative images of hematoxylin and eosin-stained liver sections 24 and 48 hr after CLP of mouse groups described in (a,b); scale bar, 50 μm. CLP 24 hr n = 9; rCD14+CLP 24 hr n = 7; CLP 48 hr n = 4, rCD14+CLP 48 hr n = 3. *p<0.05; **p<0.01; ****p<0.0001; ns, no significant difference (p>0.05); Mann-Whitney test for (e) and Log-rank (Mantel-Cox) test for (a) and (b).

The online version of this article includes the following source data and figure supplement(s) for figure 6:

**Source data 1.** Source data file for *Figure 6*.
**Figure supplement 1.** Impact of P2X7 deficiency in sepsis.
**Figure supplement 2.** Impact of P2X7 antagonist A438079 in sepsis.
**Figure supplement 3.** Impact of recombinant CD14 in sepsis.

the pro-inflammatory functions of the macrophage, which includes reduced CD14 signaling and NLRP3 inflammasome activation. However, it should be noted that the stimulation of P2X7 receptor after LPS priming enhance the release of pro-inflammatory cytokines (*de Torre-Minguela et al., 2016*; *Solle et al., 2001*).

When P2X7 receptor is absent or pharmacologically blocked, the reduced levels of circulating CD14 during sepsis is accompanied by an increase of cytokine release. Lower levels of cytokines were restored by the addition of recombinant CD14. This strongly suggests that the release of CD14 induced by P2X7 receptor during sepsis reduces the induction of cytokine by bacterial LPS, being in line with the fact that a reduced amount of CD14 at the plasma membrane impairs LPS and other PAMPs signaling (*Akashi-Takamura and Miyake, 2008*; *Baumann et al., 2010*; *Weber et al., 2012*) and circulating CD14 binds LPS and impair its signaling from plasma membrane receptors (*Kitchens and Thompson, 2005*).

CD14 release has been described during infection due to proteinase-dependent shedding; however, there is also a proteinase-independent CD14 release that is less well understood (*Wu et al., 2019*). Our study demonstrates that the release of extracellular vesicles induced by P2X7 receptor activation is a pathway contributing to the extracellular pool of CD14. During sepsis, cell-free CD14 is present in serum and other body fluids and has been proposed as a marker for septic patients (*Bas et al., 2004*; *Zhang et al., 2015*); in fact, the presence of CD14 in the blood has been validated by different studies as a valuable prognostic capacity to predict mortality (*Behnes et al., 2014*). CD14 increases in plasma during the first 24 hr after sepsis initiation and remains elevated at least during the first 8 days, conferring an exceptional long-term prognostic value over acute phase proteins or IL-6 that quickly decrease after 3–8 days of sepsis (*Behnes et al., 2014*; *Martínez-García et al., 2019*). This is similar to the CLP model presented in this study, where while serum IL-6 concentration remains constant between 1 and 2 days of sepsis initiation, CD14 increased. Extracellular CD14 is required for host defense and in particular for bacterial clearance (*Qin et al., 2017*; *Knapp et al., 2006*; *Sahay et al., 2018*; *Wieland et al., 2005*), being necessary for phagocytosis of bacteria (*Grunwald et al., 1996*; *Lingnau et al., 2007*; *Schiff et al., 1997*). Likewise, we found that P2X7 receptor deficiency or its pharmacological inhibition reduces CD14 in peritoneal lavage and serum when mice are subjected to a sepsis model. In these circumstances, there is an increase in bacterial dissemination that was controlled by the exogenous reconstitution of extracellular CD14. The decreased levels of CD14 in the infection foci of our model, the peritoneum of P2X7-deficient mice, could be then the cause of the increased to bacterial dissemination from peritoneum and infection of distant tissues and organs, thus compromising animal viability. This is in agreement with a previous study that found the P2X7 receptor to be important for bacterial clearance during sepsis (*Csóka et al., 2015*). The dissemination of bacteria in the blood during sepsis exacerbates the immune response and leads to life-threatening complications, such as organ failure and ultimately death (*Barratt-Due et al., 2017*). *P2rx7*-deficient mice present aggravated damage to different organs and premature deaths during sepsis and the administration of recombinant CD14 restores survival in the $P2rx7^{-/-}$ genotype mice. This effect is similar to wild-type mice, where the administration of CD14 increases survival (*Haziot et al., 1995*). Therefore, P2X7 receptor-dependent release of CD14 seems to have a role in bacterial infection restraint, and while we studied CD14 release from macrophages, there are also reports indicating that non-hematopoietic cells such as epithelial or endothelial cells that also express the P2X7 receptor could release CD14, thus also influencing innate immune functions during infection (*Jersmann, 2005*) and in turn restoring homeostatic conditions after sepsis (*Zanoni and Granucci, 2013*).

In conclusion, we have identified the release of CD14 by extracellular vesicles as part of the previously identified P2X7 receptor secretome of macrophages. The release of CD14 induced by the P2X7 receptor affects CD14 signaling in macrophages because the activation by smooth LPS was affected and fewer pro-inflammatory cytokines were produced. During sepsis, the elevation of CD14 levels in the serum and peritoneal lavage, also depended on the P2X7 receptor, were important in controlling cytokine secretion, restricting bacterial dissemination and organ damage, increasing overall survival. Therefore, circulating CD14 is not only a marker for sepsis but also an important component of the host's innate immune system because the P2X7 receptor releases it in a regulated manner in order to control infection and increase survival during sepsis.

# Materials and methods

## Key resources table

| Reagent type (species) or resource | Designation | Source or reference | Identifiers | Additional information |
|---|---|---|---|---|
| Strain, strain background (*S. minnesota*) | Monophosphoryl Lipid A (MPLA) | Invivogen | Cat#: tlrl-mpla | Cell culture: 1 µg/mL |
| Genetic reagent (*Mus musculus*, male) | P2RX7-deficient mice (*P2rx7-/-*) | Jackson laboratories | B6.129P2-*P2rx7tm1Gab*/J | In vivo mouse models and biological samples. RRID:IMSR_JAX:005576 |
| Antibody | Anti-MMR (rat monoclonal, clone MR5D3) | Acris antibodies | Cat#:SM1857P | WB (1:1000), RRID:AB_1611247 |
| Antibody | Anti-Cystatin B (rat monoclonal, clone 227818) | R&D Systems | Cat#: MAB1409 | WB (1:1000), RRID:AB_2086095 |
| Antibody | Anti-Cathepsin B (rat monoclonal, clone 173317) | R&D Systems | Cat#: MAB965 | WB (1:1000), RRID:AB_2086935 |
| Antibody | Anti-Peptidyl-prolyl cis-trans isomerase A (rabbit monoclonal) | Abcam | Cat#: ab41684 | WB (1:1000), RRID:AB_879768 |
| Sequence-based reagent | KiCqStart SYBR Green Primers | Sigma-Aldrich | *Tnfa* (NM_013693) *Il-6* (NM_031168) *Il1b* (NM_008361) | qRT-PCR |
| Peptide, recombinant protein | Human sCD14 recombinant protein | Preprotech | Cat#: 110–01 | In vivo: 10 µg/g RRID:AB_2877062 |
| Commercial assay, kit | ExoQuick-TC ULTRA EV isolation kit | System Biosciences (SBI) | Cat#: EQULTRA-20TC-1 | Extracellular vesicle isolation |
| Commercial assay, kit | Mouse CD14 DuoSet Elisa kit | R&D Systems | Cat#:DY982 | Detection of CD14 in biological fluids and culture supernatants. RRID:AB_2877065 |
| Commercial assay, kit | Magnetic Luminex Assay | R&D Systems | Cat#: LXSAMSM-15 | Multiplex for mice serum |
| Chemical compound, drug | ATP | Sigma-Aldrich | A2383-5G | Cell culture: 3 mM For FACS: 5 mM |
| Chemical compound, drug | A438079 | Tocris | Cat#: 2972 | Cell culture: 10–20 µM In vivo: 100 µg/kg For FACS: 10 µM |
| Software, algorithm | NTA 3.1 software | Nano Sight Technology | NS300 instrument | Nanoparticle tracking analysis, RRID:SCR_014239 |

## Mice

All experimental protocols for animal handling were refined and approved by the Animal Health Service of the General Directorate of Fishing and Farming of the Council of Murcia (*Servicio de Sanidad Animal, Dirección General de Ganadería y Pesca, Consejería de Agricultura y Agua Región de Murcia*, reference A1320140201). C57BL/6 mice (WT, wild-type, RRID:IMSR_JAX:000664) and P2X7 receptor-deficient mice in C57BL/6 background (*P2rx7*$^{-/-}$; RRID:IMSR_JAX:005576) (*Solle et al., 2001*) were obtained from the Jackson Laboratories. NLRP3-deficient (*Nlrp3*$^{-/-}$) (*Martinon et al., 2006*) and Caspase-1/11 deficient (*Casp1/11*$^{-/-}$) (*Kuida et al., 1995*) in C57BL/6 background were a generous gift from I. Coullin. For all experiments, mice between 8 and 10 weeks of age were used. Mice were bred in specific pathogen-free conditions with a 12:12 hr light-dark cycle and used in accordance with the *Hospital Clínico Universitario Virgen Arrixaca* animal experimentation guidelines, and Spanish national (Royal Decree 1201/2005 and Law 32/2007) and EU (86/609/EEC and 2010/63/EU) legislation.

## Cecal ligation and puncture

The cecal ligation and puncture (CLP)-induced sepsis procedure was performed in wild-type and *P2rx7*$^{-/-}$ mice as previously described (*Rittirsch et al., 2009*). Briefly, a laparotomy was performed to isolate the cecum of mice anesthetized with isoflurane (3–5% for induction and 1.5–2% for maintenance and oxygen flow to 1 L/min). Approximately 2/3 of the cecum was ligated with a 6–0 silk suture and punctured twice through-and-through with a 21 gauge needle. The abdominal wall and incision were closed with 6–0 silk suture. Sham operated animals underwent laparotomy without ligation or puncture of the cecum. Buprenorphine (0.3 mg/kg) was administered intraperitoneally (i.p.) at the time of surgery and mice were monitored continuously until recovery from anesthesia. Twenty-four or 48 hr after the procedure, the animals were euthanized by $CO_2$ inhalation and peritoneal lavages and blood and tissue samples were collected. In some experiments, *P2rx7*$^{-/-}$ mice received an i.p. injection of human recombinant CD14 (10 µg/g, Peprotech, RRID:AB_2877062) or vehicle (sterile physiologic saline) 30 min prior to the CLP procedure. In some experiments, wild-type mice were injected with A438079 (100 µM/kg, i.p.) or vehicle 1 hr prior to the CLP procedure.

## Mouse sample collection

Blood samples were obtained by thoracic aorta and were centrifuged at 12,500*g* for 10 min. The recovered serum was stored at −80°C until further use. For collecting peritoneal lavage, the abdominal wall was exposed by opening the skin; 4 ml of sterile saline were injected into the peritoneal cavity via a 25 gauge needle. The abdomen was gently massaged for 1 min, and the peritoneal fluid was recovered through the needle and centrifuged at 433 *g* for 10 min to obtain a cell-free peritoneal lavage. The supernatant was stored at −80°C until further analysis. For tissue harvesting, the abdominal wall was exposed, the organs were removed using scissors and forceps, and they were fixed and paraffin-embedded or stored at −80°C for future analysis.

## Quantification of bacterial colony-forming units (CFU)

Fresh liver samples were homogenized mechanically in sterile physiologic saline. Serum, peritoneal lavages, and tissue samples were diluted serially in sterile physiological saline and 100 µl of each dilution was plated in Luria-Bertani agar and cultured on agar plates at 37°C. After 24 hr of incubation, the number of bacterial colonies (CFU) was counted in the various dilutions and only used the dilutions where separate colonies were obtained. Bacterial load was calculated by multiplying CFUs to the corresponding dilution and divided by the volume inoculated to obtain the expressed CFU/ml of serum or peritoneal exudates or CFU/g of liver.

## Histopathology

Liver, spleen, and lung tissues were fixed in 4% *p*-formaldehyde (PFA, Sigma) for 24 hr, processed, paraffin-embedded, and sections stained with hematoxylin and eosin using standard methods to evaluate damage. Slides were examined using a Zeiss Axio Scope AX10 microscope with an AxioCamICC3 (Carl Zeiss).

## Differentiation and in vitro stimulation of macrophages

Bone-marrow-derived macrophages (BMDMs) were obtained from wild-type, $P2rx7^{-/-}$, $Nlrp3^{-/-}$ and $Casp1/11^{-/-}$ mice by differentiating bone marrow cells for 7 days in DMEM (Lonza) supplemented with 25% of L929 medium, 15% fetal calf serum (FCS, Life Technologies), 100 U/ml penicillin/streptomycin (Lonza), and 1% L-glutamine (Lonza) as described elsewhere (*Barberà-Cremades et al., 2012*). Cells were primed with ultrapure *E. coli* LPS serotype O55:B5 (10 ng/ml, Invivogen) or recombinant mouse IL-4 (20 ng/ml, BD Pharmingen, RRID:AB_2868873) for 4 hr. Cells were then washed three times with physiological buffer before and then stimulated for 20 min with ATP (3 mM, Sigma-Aldrich) in E-total buffer (147mMNaCl, 10 mM HEPES, 13 mM glucose, 2 mM $CaCl_2$, 1 mM $MgCl_2$, and 2mMKCl, pH 7.4). In other cases, cells were pretreated with ATP (3 mM) in the presence or absence of the specific P2X7 receptor antagonist A438079 (Tocris, 10–20 µM) in E-total buffer and then washed and stimulated with LPS or 1 µg/ml of*S. Minnesota* Monophosphoryl Lipid A (MPLA, Invivogen) for 4 hr. Then cells were treated with 10 µM of nigericin sodium salt (Sigma-Aldrich) for 30 min in E-total. In some experiments BMDMs were incubated with 20 µg/ml of the blocking antibody anti-CD14 clone M14-23 (Biolegend) before LPS or MPLA were added. Supernatants were collected and clarified at 14,000 *g* for 30 s at 4°C to remove floating cells and stored at −80°C until cytokine determination. Cells were lysed immediately in lysis buffer (50 mM Tris-HCl pH 8.0, 150 mM NaCl, 2% Triton X-100) supplemented with 100 µl/ml of protease inhibitor mixture (Sigma) for 30 min on ice and then cell debris was removed by centrifugation at 16,000 *g* for 15 min at 4°C.

## Extracellular vesicle isolation by ultracentrifugation

Extracellular vesicles were purified as previously described (*Théry et al., 2006*), diagram shown in *Figure 1—figure supplement 2c*. Briefly, differentiated BMDMs in 150 mm$^2$ plates were washed with PBS and incubated 24 hr in medium with extracellular vesicle-depleted FBS. The cells were primed with 10 ng/mL LPS, 20 ng/ml IL-4 or complete cell culture media alone for 4 hr at 37°C, then washed three times with E-total buffer and incubated in the same buffer with ATP 3 mM for 20 min. The collected medium was immediately transferred into a tube containing Protease inhibitor mix (Sigma) on ice, and then followed by sequential centrifugation at 4°C for 20 min at 2000 *g*, (Sigma 3-18KS, rotor 11180 and 13190), 30 min at 10,000 *g*, and 1 hr at 100,000 *g* (Beckman Ultracentrifuge Optima L-80 XP, SW40 rotor). The supernatant of this last step was stored at −80°C. The pellet from 100,000 *g* was washed in 10 ml of PBS and centrifuged again for 1 hr at 100,000 *g*. Finally, extracellular vesicle fraction was collected in the pellet with 50 ml of PBS and stored at −80°C until use.

## Extracellular vesicles isolation by ExoQuick-TC ULTRA

ExoQuick precipitation was carried out following the manufacturer's instructions (System Biosciences), diagram shown in *Figure 1—figure supplement 2d*. Briefly, 800 µl of cell culture supernatant or 2 ml or peritoneal lavage was diluted to 5 ml in PBS and mixed with 1 ml of ExoQuick-TC solution by inverting the tube several times. The sample was incubated overnight at 4°C then centrifuged twice at 3000 *g* for 10 min to isolate extracellular vesicles. Later, extracellular vesicles were centrifuged at 1,000 *g* for 30 s in order to purify them.

## Western blot

Cells lysates, total cell-free supernatants, extracellular vesicle fraction, and extracellular vesicle-free supernatants were resolved in 4–12% precast Criterion polyacrylamide gels (Biorad) and transferred to nitrocelulose membranes (Biorad) by electroblotting as it is described in *de Torre-Minguela et al., 2016*. Cell-free and extracellular-free supernatants were precipitated overnight at −20°C with 6 vol of cold acetone. Membranes were probed with different antibodies: anti-CD14 rat monoclonal (rmC5-3, BD Pharmingen, RRID:AB_395020), anti-CD9 rabbit monoclonal (EPR2949, ab92726, Abcam, RRID:AB_10561589), anti-MMR rat monoclonal (MR5D3, Acris Antibodies, RRID:AB_1611247), anti-Cystatin B rat monoclonal (Clone #227818, R and D, RRID:AB_2086095), anti-Cathepsin B rat monoclonal (Clone #173317, R and D, RRID:AB_2086935), or anti-Peptidyl-prolyl cis-trans isomerase A rabbit polyclonal (ab41684, Abcam, RRID:AB_879768).

## Nanoparticle tracking analysis

After ultracentrifugation, 100K pellet was analyzed with an NS300 nanoparticle tracking analysis (NTA) instrument, (NanoSight Technology) to determine the vesicle size distributions and concentrations. Data was analyzed with NTA 3.1 software (RRID:SCR_014239).

## Transmission electron microscopy (TEM)

Electron microscopy analysis was performed as previously described (*Théry et al., 2006*) on pellets of purified extracellular vesicle loaded on form var-carbon-coated grids and fixed in 2% PFA. Grids were observed at 80 kV with a JEM-1011 Transmission Electron Microscope (JEOL Company). EV were counted for each preparation in five different random fields of TEM pictures taken at 25,000x. The number of EV was then normalized to the number of cells obtained in each treatment.

## Flow cytometry

For membrane CD14 flow cytometry, BMDMs seeded in 24-well plates were washed and incubated for 30 min at 37°C in E-total buffer supplemented with or without 5 mM of ATP, in presence or absence of P2X7 receptor antagonist A438079 (10 µM). To stain surface CD14, cells were washed and incubated with mouse seroblock FcR (BD biosciences) and then stained with anti-mouse CD14 (clone rmC5-3; 553738; BD biosciences; RRID:AB_395020) for 30 min at 4°C. Cells were washed again and incubated with secondary Alexa Fluor 647 goat anti-rat IgG (H+L) (A21247; Invitrogen, RRID:AB_141778) for an additional 30 min at 4°C. Finally, cells were washed and fixed with 4% PFA in PBS and then scrapped and aliquoted in flow cytometry tubes. For human P2X7 flow cytometry, monocytes were determined from peripheral blood mononuclear cells from non-septic and septic patients by CD3$^-$ CD14$^+$ selection, and P2X7 receptor surface expression was determined using the monoclonal anti-P2X7 L4 clone (*Buell et al., 1998*; *Martínez-García et al., 2019*). All samples were subjected to flow cytometry analysis using a BD FACSCanto flow cytometer (BD) and FACSDiva software (BD, RRID:SCR_001456) by gating for BMDM cells based on FSC versus SSC parameters.

## Quantitative reverse transcriptase–polymerase chain reaction (RT-PCR) analysis

BMDMs, plated in 96-well plates, were stimulated as described above. Total RNA extraction was performed using the RNAqueous Micro Kit (Invitrogen), followed by reverse transcription using iScript cDNA Synthesis (Bio-Rad) with oligo-dT. The mix SYBR Premix ExTaq (Takara) was used for quantitative PCR in iCycler My iQ thermocycler (Bio-Rad). Specific primers were purchased from Sigma (KiCq Start SYBR Green Primers). Only a single product was seen on melting curve analysis, and for each primer set, the efficiency was >95%. For the relative expression of mouse *Il6*, *Tnfa*, and *Il1b*, their Ct was normalized to the housekeeping gene *Gapdh* using the $2^{-\Delta Ct}$ method.

## Human clinical samples

The samples and data from patients included in this study were provided by the *Biobanco en Red de la Región de Murcia* (PT13/0010/0018), which is integrated into the Spanish National Biobanks Network (B.000859) and approved by the clinical ethics committee of the Clinical University Hospital *Virgen de la Arrixaca* (reference numbers PI13/00174, 2019-9-4-HCUVA, 2019-12-15-HCUVA and 2019-12-14-HCUVA). All study procedures were conducted in accordance with the declaration of Helsinki. Whole peripheral blood samples were collected after receiving written informed consent from intraabdominal sepsis patients (*n* = 9, *Supplementary file 1*-Table 1) at the Surgical Critical Unit from the Clinical University Hospital *Virgen de la Arrixaca*. The blood samples were obtained from septic individuals within 24 hr of the diagnosis of sepsis. The inclusion criteria for septic patients were patients diagnosed with intra-abdominal origin sepsis confirmed by exploratory laparotomy, with at least two diagnostic criteria for sepsis (fever or hypothermia; heart rate >90 beats per minute; tachypnea, leukocytosis, or leukopenia) and multiple organ dysfunction defined as physiological dysfunction in two or more organs or organ systems (*Singer et al., 2016*). We also recruited non-septic volunteers and after they had signed their informed consent agreement whole peripheral blood samples were collected (*n* = 10). Sera was isolated and stored at −80°C until use.

## ELISA and multiplex assay

Individual culture cell-free supernatants were collected and clarified by centrifugation. The concentration of IL-6 (RRID:AB_2877063), TNF-$\alpha$ (RRID:AB_2877064), IL-1$\beta$ (RRID:AB_2574946), and CD14 (RRID:AB_2877065) was tested by ELISA following the manufacturer's instructions (R and D Systems and Thermo Fisher). Mice serum and peritoneal lavages were collected and the concentration of IL-6 and CD14 was also tested by ELISA (R and D Systems). Results were read in a Synergy Mx (BioTek) plate reader. Multiplexing in mice serum for MCSF, CRP, RAGE, Resistin, VEGF, MIP-1$\alpha$, MIP-1$\beta$, MIP-2, CCL2, CCL5, IL-1$\alpha$, IL-5 and IL-10was performed using the Luminex color-coded superparamagnetic beads array from R and D Systems following the manufacturer indications, and the results were analyzed in a Bio-Rad Bio-Plex analyzer.

## Statistical analysis

Statistical analyses were performed using GraphPad Prism 7 (Graph-Pad Software, Inc, RRID:SCR_002798). For two-group comparisons, the Mann-Whitney test was used and Kaplan-Meier survival curves were plotted and the log-rank test was undertaken to determine the statistical significance. The $\chi^2$ test was used to determine whether there was a significant difference between different clinical variables among groups of septic patients, except for age, where a one-way ANOVA test was used. For mouse in vivo data and before statistical analysis, possible outliers were identified with the robust regression followed by outlier identification method with Q = 1% and were eliminated from the analysis and representation. All data are shown as mean values and error bars represent standard error from the number of independent assays indicated in the figure legend. p Value is indicated as *p<0.05; **p<0.01; ***p<0.001; ****p<0.0001; p>0.05 not significant (*ns*).

# Acknowledgements

We thank MC Baños and AI Gómez (IMIB-Arrixaca, Murcia, Spain) for technical assistance with molecular and cellular biology and the members of Dr. Pelegrin's laboratory for comments and suggestions, especially to Dr. M Pizzuto for critical reading the manuscript. We also acknowledge the patients and volunteers who enrolled in this study, the BioBank Biobanco en Red de la Región de Murcia (PT13/0010/0018), which is integrated into the Spanish National Biobanks Network (B.000859) for its collaboration and the SPF-animal house from IMIB-Arrixaca for mice colony maintenance.

# Additional information

## Funding

| Funder | Grant reference number | Author |
|---|---|---|
| FEDER/Ministerio de Ciencia, Innovación y Universidades - Agencia Estatal de Investigación | SAF2017-88276-R | Pablo Pelegrin |
| European Research Council | 614578 | Pablo Pelegrin |
| European Research Council | 899636 | Carlos García-Palenciano Pablo Pelegrin |
| Fundación Séneca | 21081/PDC/19 | Pablo Pelegrin |
| Fundación Séneca | 20859/PI/18 | Carlos García-Palenciano Pablo Pelegrin |

The funders had no role in study design, data collection and interpretation, or the decision to submit the work for publication.

## Author contributions

Cristina Alarcón-Vila, Conceptualization, Formal analysis, Investigation, Methodology, Writing - original draft, Writing - review and editing; Alberto Baroja-Mazo, Carlos de Torre-Minguela,

Conceptualization, Formal analysis, Investigation, Writing - original draft, Writing - review and editing; Carlos M Martínez, Formal analysis, Investigation, Writing - review and editing; Juan J Martínez-García, Helios Martínez-Banaclocha, Investigation, Writing - review and editing; Carlos García-Palenciano, Conceptualization, Investigation, Writing - review and editing; Pablo Pelegrin, Conceptualization, Formal analysis, Supervision, Funding acquisition, Writing - original draft, Writing - review and editing

### Author ORCIDs
Carlos de Torre-Minguela (iD) https://orcid.org/0000-0003-3314-3203
Carlos M Martínez (iD) https://orcid.org/0000-0003-3307-1326
Pablo Pelegrin (iD) https://orcid.org/0000-0002-9688-1804

### Ethics
Human subjects: The samples and data from patients included in this study were provided by the Biobanco en Red de la Región de Murcia (PT13/0010/0018), which is integrated into the Spanish National Biobanks Network (B.000859) and approved by the clinical ethics committee of the Clinical University Hospital Virgen de la Arrixaca (reference numbers PI13/00174, 2019-9-4-HCUVA, 2019-12-15-HCUVA and 2019-12- 471 14-HCUVA). All study procedures were conducted in accordance with the declaration of Helsinki. Whole peripheral blood samples were collected after receiving written informed consent from intraabdominal sepsis patients at the Surgical Critical Unit from the Clinical University Hospital Virgen de la Arrixaca.

Animal experimentation: All experimental protocols for animal handling were refined and approved by the Animal Health Service of the General Directorate of Fishing and Farming of the Council of Murcia (Servicio de Sanidad Animal, Dirección General de Ganadería y Pesca, Consejería de Agricultura y Agua Región de Murcia, permit reference A1320140201). Mice were used in accordance with the Hospital Clínico Universitario Virgen Arrixaca animal experimentation guidelines (Permit Number 221116/1/PE), and Spanish national (Royal Decree 1201/2005 and Law 32/2007) and EU (86/609/EEC and 335 2010/63/EU) legislation. All surgery was performed under sodium pentobarbital anesthesia, and every effort was made to minimize suffering.

### Decision letter and Author response
Decision letter https://doi.org/10.7554/eLife.60849.sa1
Author response https://doi.org/10.7554/eLife.60849.sa2

## Additional files

### Supplementary files
• Supplementary file 1. Demographics and clinical features of enrolled healthy volunteers and patients with intra-abdominal sepsis. Table 2. Histopathology scoring (average of *n* = 3 animals/group).

• Transparent reporting form

### Data availability
All data generated or analysed during this study are included in the manuscript and provided as raw data as single values for all Figures.

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
