## [Decision Letter]

Thank you for submitting your article "CD14 release induced by P2X7 receptor restrict inflammation and increases survival during sepsis" for consideration by *eLife*. Your article has been reviewed by three peer reviewers, including Evangelos J Giamarellos-Bourboulis as the Reviewing Editor and Reviewer #1, and the evaluation has been overseen by Tadatsugu Taniguchi as the Senior Editor. The following individual involved in review of your submission has agreed to reveal their identity: Sebastian Weis (Reviewer #2).

The reviewers have discussed the reviews with one another and the Reviewing Editor has drafted this decision to help you prepare a revised submission.

All three reviewers agreed that your manuscript includes novel data that establish a causal link between P2X7 receptor signaling, the release of CD14 and sepsis outcome. I attach the comments of the reviewers and I ask that you particularly revise and perform the additional experiments asked by the third reviewer (third comment).

Reviewer #1:

This is a fascinating manuscript on the interaction of CD14 and the P2X7 receptor to modulate survival and bacterial growth in sepsis. I have some points of concerns:

• The clinical data does not offer much, unless the authors have evidence of the gene expression of P2X7. I suggest that either the authors provide such data or that they omit all clinical information from the manuscript.

• The Discussion should be enriched with literature on the kinetics of sCD14 in human sepsis.

• The authors need to better address how they achieved quantitative bacterial culture of 6log10 in serum (Figure 5). What type of growth medium did they use to inoculate?

• What types of bacteria were isolated in tissues and peritoneal lavage? Is there some specificity for the species to be favoured in the absence of P2X7? Is that species associated with inflammasome activation?

Reviewer #2:

Alarcón-Vila et al. investigate the role of CD14 in macrophage function and during polymicrobial sepsis. Using in vitro and in vivo models, including the gold standard animal model CLP, as well as data from sepsis patient, they here establish a causal link between P2X7 receptor signaling, the release of CD14 and sepsis outcome. The data is novel and interesting.

The manuscript is very well written, it is build up in a logical way and is easy to follow. The presented data is very convincing and clearly match the statements made in the manuscript.

Reviewer #3:

With interest, I reviewed the manuscript by Alarcón-Vila et al., who investigated the role of the P2X7 receptor in the release of CD14 in extracellular vesicles and its role in the septic immune response and bacterial dissemination. This work appears to tie several of these previous findings together in the context of sepsis. However, I do have several important issues, especially regarding the novelty of this study and the congruency of the present results with previous findings, that preclude publication of this work in its present form.

1) Many findings of the current work have been shown before. For instance, the same n group has reported LPS/ATP-induced extracellular release of CD14 in BMDMs (de Torre-Minguela et al., 2016). The authors need to clearly highlight which findings of the current work are novel.

2) The authors report increased cytokine (TNF, IL-6) release in P2rx7- LPS/ATP-treated BMDMs compared with WT BMDMs (Figure 2A, B). In agreement, they demonstrate increased serum pro-inflammatory cytokine levels in P2rx7- mice subjected to CLP compared with WT CLP mice. However, these results appear to be in contrast to previous data from their own group (de Torre-Minguela et al., 2016), who report less cytokine production in P2rx7- BMDMs stimulated with LPS/ATP. Furthermore, Solle et al., 2001, also report less cytokine production in P2rx7- mice challenged with LPS/ATP compared with WT mice. How can the authors reconcile their current findings with these previous studies.

3) In Figure 1B, the authors show that the LPS/ATP-induced release of extracellular CD14 by BMDMs can be counteracted by a P2X7 receptor antagonist. The authors should perform the same experiment in P2rx7- BMDMs to confirm that CD14 release is truly p2Rx7-dependent. They clearly have access to these cells (see Figures 2-6).

4) How do the authors reconcile the findings in Figure 2F (coincubation with ATP strongly reduces LPS-induced IL-1β production by BMDMs) with those of Netea et al. (Blood 2009), who demonstrate that LPS alone is not sufficient for robust IL-1β production by macrophages, but that this response necessitates costimulation with ATP?

5) In Figure 3B, there is still a lot of CD14 shedding in P2rx7- mice, what is the alternative mechanism?

---

## [Author Response]

Reviewer #1:This is a fascinating manuscript on the interaction of CD14 and the P2X7 receptor to modulate survival and bacterial growth in sepsis. I have some points of concerns:• The clinical data does not offer much, unless the authors have evidence of the gene expression of P2X7. I suggest that either the authors provide such data or that they omit all clinical information from the manuscript.

We appreciate reviewer comment, we do now include new data on the expression of P2X7 receptor on the surface of monocytes stained by flow cytometry using the L4 mAb. P2X7 receptor presents an increase in surface expression in septic patients. We feel that this data is better than gene expression, since the mRNA expression of P2X7 receptor could result in channels not traduced or trafficked to plasma membrane and therefore, in non-functional channels. New data is shown in Figure 3A and text:

“Similarly, the presence of P2X7 receptor in monocytes was also elevated in septic individuals (Figure 3A), in accordance to previous studies (Martínez-García et al., 2019).”

We also describe the method in the subsection “Flow cytometry”.

• The Discussion should be enriched with literature on the kinetics of sCD14 in human sepsis.

We have now increased the Discussion with references about sCD14 in human sepsis, as in the last few years, the relevance of circulating CD14 in human sepsis has been widely investigated. See Discussion:

“During sepsis, cell-free CD14 is present in serum and other body fluids and has been proposed as a marker for septic patients (Bas et al., 2004; Zhang et al., 2015), in fact the presence of CD14 in the blood has been validated by different studies as a valuable prognostic capacity to predict mortality (Behnes et al., 2014). […] This is similar to the CLP model presented in this study, where while serum IL-6 concentration remains constant between 1 and 2 days of sepsis initiation, CD14 increased.”

• The authors need to better address how they achieved quantitative bacterial culture of 6log10 in serum (Figure 5). What type of growth medium did they use to inoculate?

We are not sure to what the reviewer is referring, since in Figure 5 we do not present log_10_ data. We dispense 100µL (of different serum dilutions) in Luria-Bertani (LB) agar (growth medium) and we distributed the sample with a sterile Drigalski spatula until the inoculum is fully dispersed on the agar surface. Plates are incubated (37ºC for 24h) until colony forming units (CFUs) were evident. We counted the CFUs in the various dilutions and we only use the plates in which separate colonies were obtained (maximum of 200 CFUs/plate). Then we calculated the bacterial load by multiplying CFUs to the corresponding dilution and divided by the volume inoculated to obtain CFUs/mL. We have now added a better description to the Materials and methods subsection “Quantification of bacterial colony forming units (CFU)”.

• What types of bacteria were isolated in tissues and peritoneal lavage? Is there some specificity for the species to be favoured in the absence of P2X7? Is that species associated with inflammasome activation?

This is an interesting observation, but unfortunately, we did not analyze the type of bacteria isolated in the tissues during the sepsis model and we are unable to provide such values. In base to the literature the CLP model induces the dissemination of different types of enteric bacteria (mainly *Escherichia coli*) and we now present data of the type of bacteria isolated from the blood of intra-abdominal sepsis origin patients included in this study (see updated Supplementary file 1—table 1).

Reviewer #3:With interest, I reviewed the manuscript by Alarcón-Vila et al., who investigated the role of the P2X7 receptor in the release of CD14 in extracellular vesicles and its role in the septic immune response and bacterial dissemination. This work appears to tie several of these previous findings together in the context of sepsis. However, I do have several important issues, especially regarding the novelty of this study and the congruency of the present results with previous findings, that preclude publication of this work in its present form.1) Many findings of the current work have been shown before. For instance, the same n group has reported LPS/ATP-induced extracellular release of CD14 in BMDMs (de Torre-Minguela et al., 2016). The authors need to clearly highlight which findings of the current work are novel.

We agree that we have previously reported release of CD14 from P2X7 activated macrophages, however in this current work we go further and find that CD14 is released by extracellular vesicles from ATP-treated macrophages, we also show the functional consequences of CD14 release from macrophages (i.e. decrease in LPSactivation) and furthermore, we also depict the in vivo role of released CD14 promoting sepsis survival (decreasing bacterial load and cytokine storm). Moreover, we also found that the treatment of mice with soluble recombinant CD14 resulted in increase of survival, opening the possibility of new therapeutic avenues for sepsis. We have better highlighted this in the Abstract:

“Here we show for first time that P2X7 receptor induces the release of CD14 in extracellular vesicles, resulting in a net reduction in macrophage plasma membrane CD14 that functionally affects LPS, but not monophosphoryl lipid A, pro-inflammatory cytokine production.”

End of Introduction:

“In this study, we found for first time that CD14 is a cargo of extracellular vesicles released by P2X7 receptor activation and functionally that the lack of cellular CD14 compromises the production of macrophage pro-inflammatory cytokines. Additionally, we also found that during sepsis there is a decrease in the extracellular pool of CD14 in P2rx7−/− mice”

and at the beginning of the Discussion:

“Our study shows for first time that the cellular release of CD14 induced by the P2X7 receptor has two functional effects on the innate immune system”

2) The authors report increased cytokine (TNF, IL-6) release in P2rx7- LPS/ATP-treated BMDMs compared with WT BMDMs (Figure 2A, B). In agreement, they demonstrate increased serum pro-inflammatory cytokine levels in P2rx7- mice subjected to CLP compared with WT CLP mice. However, these results appear to be in contrast to previous data from their own group (de Torre-Minguela et al., 2016), who report less cytokine production in P2rx7- BMDMs stimulated with LPS/ATP. Furthermore, Solle et al., 2001, also report less cytokine production in P2rx7- mice challenged with LPS/ATP compared with WT mice. How can the authors reconcile their current findings with these previous studies.

We appreciate the point raised by the reviewer, in Figure 2A and B, we found that both treatments with LPS alone or stimulation with ATP and then with LPS resulted in increased TNFa and IL-6 production as the reviewer point out. We would like to highlight and clarify the point of this study, as in Figure 2A, B ATP was added before LPS, in contraposition with our own previous study (de Torre-Minguela et al., 2016) as well as Solle et al., 2001, where ATP was added **after** LPS priming. When P2X7 receptor is triggered by ATP before LPS, then CD14 shedding impairs optimal LPS priming, this is reverted in P2X7-KO macrophages, that eventually, potential autocrine P2X7 activation during cellular plating and previous to LPS application, could be slightly impairing cytokine production. We have indicated this peculiarity in the different panels of the Figure 2, as “ATP (pre LPS)”, and better explain this in the modified manuscript:

“Treatment of macrophages with extracellular ATP and then subjected to LPS activation resulted in a decrease in LPS-induced expression and a secretion of the proinflammatory cytokines IL-6 and TNF-α”

“The reduction in cytokine production upon LPS-priming induced by initial P2X7 receptor activation in macrophages is additional to the effect we have described on the inflammasome activation (Martínez-García et al., 2019),”

“However, it should be noted that the stimulation of P2X7 receptor after LPS priming enhance the release of pro-inflammatory cytokines (de Torre-Minguela et al., 2016; Solle et al., 2001).”

3) In Figure 1B, the authors show that the LPS/ATP-induced release of extracellular CD14 by BMDMs can be counteracted by a P2X7 receptor antagonist. The authors should perform the same experiment in P2rx7- BMDMs to confirm that CD14 release is truly p2Rx7-dependent. They clearly have access to these cells (see Figures 2-6).

As suggested, we have now confirmed data obtained with P2X7 receptor antagonist using P2X7-KO macrophages. Please see new data on Figure 1B.

4) How do the authors reconcile the findings in Figure 2F (coincubation with ATP strongly reduces LPS-induced IL-1β production by BMDMs) with those of Netea et al. (Blood 2009), who demonstrate that LPS alone is not sufficient for robust IL-1β production by macrophages, but that this response necessitates costimulation with ATP?

We are sorry for this misinterpretation, in Figure 2F the release of IL-1b was induced by nigericin treatment after LPS, this was specified in the figure legend but not in the histogram. Now we indicate this in the Figure 2F histogram to avoid misinterpretation.

Nigericin is a K^+^ ionophore widely used to activate the NLRP3 inflammasome (for review about nigericin induced NLRP3 see i.e. Próchnicki et al., F1000Research PMID: 27508077). Therefore, our data are in agreement with Netea et al., 2009 and many other studies (including our previous publications), where LPS alone does not induce IL-1b release, instead it requires a two-step model: first an LPS priming and then a second step with nigericin (or ATP) to induce NLRP3 activation.

5) In Figure 3B, there is still a lot of CD14 shedding in P2rx7- mice, what is the alternative mechanism?

During the complex scenario of the CLP model, there are multiple different pathways triggered besides P2X7 receptor activation, it is normal to expect that some of these pathways could be also contributing to extracellular CD14 in biological fluids as serum and in the peritoneal cavity. In fact, proteinase-dependent CD14 release has been described during infection (Wu et al., 2019). However, since the ‘90s other proteinase-independent pathways for circulating CD14 are known to exist but never described in detail (Wright et al., 1990, PMID: 1698311). Our study demonstrates for first time that the P2X7 receptor is one pathway responsible to induce extracellular CD14 in vivo during infection. This was commented in the Discussion:

“CD14 release has been described during infection due to proteinase dependent shedding; however, there is also a proteinase-independent CD14 release that is less well understood (Wu et al., 2019). Our study demonstrates that the release of extracellular vesicles induced by P2X7 receptor activation is a pathway contributing to the extracellular pool of CD14.”